# A New Approach for Detecting Sublethal Effects of Neonicotinoids on Bumblebees Using Optical Sensor Technology

**DOI:** 10.3390/insects14080713

**Published:** 2023-08-17

**Authors:** Vasileia Chatzaki, Marta Montoro, Rámi El-Rashid, Annette Bruun Jensen, Antoine Lecocq

**Affiliations:** 1Department of Plant and Environmental Sciences—PLEN, University of Copenhagen, Thorvaldsensvej 40, 1871 Frederiksberg, Denmark; abj@plen.ku.dk (A.B.J.); antoine@plen.ku.dk (A.L.); 2FaunaPhotonics APS, Støberigade 14, 2450 Copenhagen, Denmark; m.montoro.caceres@gmail.com (M.M.); rami.elrashid@gmail.com (R.E.-R.)

**Keywords:** bumblebees, insect monitoring, pesticide, acute exposure, chronic exposure, infrared sensor, convolutional neural networks

## Abstract

**Simple Summary:**

Several lab studies have shown that pesticides are causing sublethal effects on bees; however, little is known about their impact in actual field conditions. Continuous surveillance of the bees in the field will help monitor their pesticide exposure and better understand their impact on bee health. This study investigated the potential use of a new optical sensor, coupled with machine learning, to automatically identify flying insects, as an approach to monitoring the sublethal effects of pesticides on insects. To do so, bumblebees, *Bombus terrestris*, were exposed to field-realistic doses of a neonicotinoid pesticide, and their flight activity was recorded by the sensor. Through the data acquired, the investigation focused on whether the machine learning algorithm could distinguish the flight events that were created by the healthy bumblebees and the bumblebees that had been exposed to sublethal doses of the pesticide. The results showed that the algorithm could differentiate flight events, indicating the promise of the sensor as a continuous monitoring tool for bee health in the field.

**Abstract:**

Among insects, bees are important pollinators, providing many vital ecosystem services. The recent pollinator decline is threatening both their diversity and abundance. One of the main drivers of this decline is the extensive use of pesticides. Neonicotinoids, one of the most popular groups of pesticides, can be toxic to bees. In fact, numerous studies have found that neonicotinoids can cause sublethal effects, which can impair the biology, physiology, and colony survival of the bees. Yet, there are still knowledge gaps, and more research is needed to better understand the interaction between neonicotinoids and bees, especially in the field. A new optical sensor, which can automatically identify flying insects using machine learning, has been created to continuously monitor insect activity in the field. This study investigated the potential use of this sensor as a tool for monitoring the sublethal effects of pesticides on bumblebees. *Bombus terrestris* workers were orally exposed to field-realistic doses of imidacloprid. Two types of exposures were tested: acute and chronic. The flight activity of pesticide-exposed and non-exposed bumblebees was recorded, and the events of the insect flights recorded by the sensor were used in two ways: to extract the values of the wingbeat frequency and to train machine learning models. The results showed that the trained model was able to recognize differences between the events created by pesticide-exposed bumblebees and the control bumblebees. This study demonstrates the possibility of the optical sensor for use as a tool to monitor bees that have been exposed to sublethal doses of pesticides. The optical sensor can provide data that could be helpful in managing and, ideally, mitigating the decline of pollinators from one of their most major threats, pesticides.

## 1. Introduction

Bumblebees are highly adapted species able to survive in diverse climates and habitats [1]. Their subfamily contains about 260 species that are mainly distributed in the Holarctic, but some species are also found in the Oriental and Neotropical regions [1]. Bumblebees are generalist pollinators, indiscriminately pollinating most wild plants and crops [2] and have the ability to visit many flowers per minute [3]. Bumblebee behaviour and features make them important and efficient pollinators [2]. In fact, some of the most common bumblebee species, such as *Bombus terrestris* (Linnaeus, 1758) and *Bombus lucorum* (Linnaeus, 1761), increase the yield value by approximately USD 425 per hectare [4]. There is a current decline in pollinator populations in many parts of the world [5]. The cause of pollinator decline is complex because its drivers are diverse and interconnected. One of the most important drivers of their decline is the intensive use of pesticides [6]. To protect pollinators, many regulatory authorities require pesticides to be tested for potential side effects before they become available on the market. These tests usually follow standard risk assessment protocols, which are mainly designed for honeybee workers (*Apis melifera*) (Linnaeus, 1758), testing acute toxicity and determining the median lethal doses (LD50) or lethal concentrations (LC50) in lab conditions [7,8,9]. However, several studies have shown that pesticide toxicity varies across species [10,11]. Considering the findings, a test guideline for the assessment of acute oral toxicity of pesticides under lab conditions, especially for *Bombus* spp., was created [12]. More specifically, the guidelines provide detailed instructions regarding the specific conditions, substance preparation, test design, exposure duration, observation parameters, and data collection that should be followed to cater to the specific requirements of *Bombus terrestris*. The aim is to standardize the testing procedure for evaluating the acute contact toxicity of chemicals on *Bombus terrestris*, ensuring consistent and comparable results among various studies and laboratories [12].

One of the most widely used pesticide groups is the neonicotinoids, which has raised several concerns regarding their undesirable effects on pollinators [13,14,15,16]. Neonicotinoids are neurotoxic compounds that block nerve impulses in the central nervous system of insects, causing paralysis [17]. They are systemic insecticides which are applied to plants as foliar sprays and to soil as drenches, in granular form, or as seed coating. Seed coating is the most common application method of neonicotinoids as their systemic action offers protection to the plant as it grows [15,17]. However, only around 10% is absorbed by the plant while the rest remains in the soil [15]. Neonicotinoids are highly persistent in the soil and accumulate in the environment [17]. They are also water soluble, spreading beyond the zones of application through bodies of water into surrounding areas [17]. Therefore, untreated plants may absorb neonicotinoid residues in areas where they were never applied and become sources of exposure [18,19].

Several studies have investigated the lethal and sublethal risks of neonicotinoids in pollinators [20,21,22]. The results of these studies, together with the concerns of beekeepers in Italy [23] and France [24,25], motivated the EU, in 2013, to introduce a moratorium on the use of three neonicotinoids, imidacloprid, clothianidin, and thiamethoxam, for outdoor use in bee-attractive crops and requested further research [26]. After further evaluation, the EU expanded the ban on the three substances to all field crops, with their use permitted only in permanent greenhouses [27]. A study that compared the exposure of bumblebees in rural areas before and after the EU moratorium found that their exposure to imidacloprid declined after the EU ban [28]. However, a study that was conducted in Sweden after the total ban on the outdoor use of neonicotinoids found that imidacloprid residues, coming from contaminated discharges of greenhouses, were present in wild plants growing along the streams [29]. Nonetheless, the EU ban decision raised concerns for economic and environmental costs. Since the moratorium, crop yields have decreased, while pest control necessities have increased, and neonicotinoids have been replaced with other insecticides which are not as effective but can still harm pollinators [24,30]. Therefore, in 2020, France decided the derogation of the neonicotinoid ban and permitted the use of sugar beet seeds treated with imidacloprid or thiamethoxam to fight the beet yellow virus (BYMY), which is transmitted by aphids [31]. 

Camp and Lehmann (2021) reviewed 78 primary research studies regarding neonicotinoids and *Bombus* spp. [32]. Most of these studies exposed bumblebees to field-realistic concentrations of different neonicotinoid compounds and investigated their sublethal effects. The field-realistic concentration of imidacloprid found in nectar is 10 ppb [33,34] which means that, given the fact that a bumblebee worker consumes in a full day 150–300 μL of nectar, a worker ingests 1,5–3 ng of imidacloprid/day [35]. The findings of these studies show that exposure to field-realistic concentrations of neonicotinoids can impair their physiology [36,37,38], behaviour [20,35,37,39,40,41,42], and colony survival [22,43,44,45]. Among all of these studies, there are some that focus on the sublethal effects of field-realistic concentration of neonicotinoids on bumblebee flight and forage activity. For example, one study proved that both acute and chronic exposure to imidacloprid decreased the pollen foraging efficiency of bumblebees [40] while another study found that chronic exposure to thiamethoxam has an impact on foraging and homing success for bumblebees [45]. Kenna et al. (2019) was the first study that tested how the flight dynamics of *Bombus* spp. are affected by neonicotinoid exposure [41]. Pesticide-treated bumblebees flew with higher velocity in a flight mill experiment; however, they terminated their flight earlier compared to the control bumblebees, and they flew a shorter distance [41]. Additionally, bumblebees that were foraging in an artificial floral array did so in a suboptimal way by flying further distances to forage and thus consuming more energy when they were exposed to an acute dose of imidacloprid [42].

Most of the aforementioned studies were conducted under laboratory conditions; therefore, to have a more complete picture of the exposure effects of neonicotinoids on bees more research is needed especially for field exposure [32,46]. The development of novel monitoring methods could thus provide important field data and expand our understanding of the impact of pesticides on bees [14,47]. In the field, insect monitoring can be carried out using conventional methods like traps, sweep nets, baits, and aspirators [48]. Remote automated insect detection tools, like acoustic detectors [49], automatic traps [50] and cameras [51], radars [52,53], lidar [54], and optical sensors [55], have also been considered. The optical sensor described in Rydhmer et al. (2022) detects flying insect activity in a given area by emitting infrared light and recording the backscattered signal from insects flying in the measurement volume [55]. The recorded signals generate insect data which are analysed using different methods, by evaluating features of the insect’s signal (for example the wingbeat frequency, body size, wing size) or by machine learning. The underlying theory is that the recorded signals differ, and every insect species has its own insect signal due to its morphological features and flight pattern behaviour [56]. The analysis is used in a multitude of commercial cases, such as categorization of insects into groups like pests, beneficials, and pollinators, or monitoring the migration of a specific pest [56]. The optical sensor is a real-time, automated, fast, and cost-effective tool for monitoring insect activity [55]. This study attempted to evaluate whether the above-described optical sensor [55] could detect any differences in the flight activity between pesticide-exposed and non-exposed (control) bumblebees. Our results support the potential use of optical sensors as a continuous monitoring tool in the field to assess the presence of pesticide-exposed bees. The acquired data could, in the future, be helpful to effectively manage and, ideally, mitigate the decline in population and diversity of bees and other pollinators. 

## 2. Materials and Methods

To investigate if the sensor can detect any sublethal effects of imidacloprid on bumblebees, experiments were conducted under laboratory conditions where the flight activity was recorded inside specially designed cages. Bumblebees were orally exposed to a field-realistic concentration of imidacloprid, which corresponds to 10μg/L (10 ppb) [46,57], and chronic and acute exposure were tested.

### 2.1. Handling of Bumblebees 

The species selected for this study was *B. terrestris*, the European buff-tailed bumblebee. Four hives of *B. terrestris* were obtained from Biobest (Westerlo, Belgium). Three of them were standard hives containing more than 80 workers and one was a premium hive containing more than 110 workers. After their arrival, the hives were kept in the lab under standard conditions (25 °C) and were fed with the artificial nectar included in the hive box (Biogluc^®^) until the start of the experiments. Each experiment was conducted one or two days after receiving each hive. For each repetition, bumblebees were collected from different hives, and from each hive, individuals of different sizes were sampled to avoid size-biased data since wing beat frequency (wbf) can be affected by the size of the body [58]. Newly emerged workers were excluded from the experiments as proposed by the OECD [12].

### 2.2. Pesticide Preparation

First, a stock solution of imidacloprid (C9H10ClN5O2 powder; grade: PESTANAL^®^, analytical standard) was prepared by dissolving 10 mg of imidacloprid powder in 100 mL of distilled water. The stock solution was covered with aluminium foil to prevent UV degradation [59] and was kept in the fridge at 5 °C. For the acute exposure experiment, 3 mL of the stock solution was diluted in 1 L of 50% (*w*/*v*) sugar water. Every bee was fed with 10 μL of the resulting concentration, thus consuming 3 ng of imidacloprid, which corresponds to the field-realistic dose (10 ppb) [35]. For the chronic exposure experiment, an aliquot of 100 μL from the stock solution was diluted in 1 L of 50% (*w*/*v*) sugar water to produce a field-realistic concentration of 10 μg/L [57]. For the control groups, both in the acute and chronic exposure experiments, a 50% (*w*/*v*) sugar water solution was prepared. All solutions were prepared before the start of each experiment’s repetition.

### 2.3. Flight Cages with Sensor Setup and Flight Recording

Two specially designed cages equipped with an optical sensor [55] were used for recording the flight activity. The cage size was approximately 1 m^3^; this size allowed every insect flight happening inside the cage to be recorded by the sensor. The flight cage was made of black fabric to absorb reflections created by the light of the sensor except for its top part, which was made from a white mesh so that natural light could enter the cage. Both cages were installed in the same lab room in order to achieve identical environmental conditions inside both cages, because wbf can be affected by temperature and humidity [60]. To keep constant tracking of the climatic conditions, an environmental sensor was installed inside both cages. The optical sensors were powered via a current from a DC power supply and constantly registered any flight activity. During the experiments, the data were sent to the cloud via mobile data, using a SIM card. For both experiments and their repetitions, the same cage was used for the pesticide-treated bumblebees to avoid contamination with pesticide in the control treatment. After the end of every experiment, the bumblebees were removed, and the cages were cleaned. Inside the cage, bumblebees had access to ad libitum 50% (*w*/*v*) sugar water solution. The sugar water was provided inside 30 mL cups filled with cotton on the top and hung from the cages’ walls to induce flight activity. The cups were refilled every day with fresh sugar water. After the recording of insect events, the bumblebees were removed from the cages, the session was terminated, and the experiment ended.

### 2.4. Acute Exposure Experiment

First, the bumblebee hive was placed under red light, where bumblebee workers were carefully collected from the hive with forceps. For every treatment, approximately 50–70 bumblebees were used (depending on the repetition). Each bumblebee was placed into a 15 mL falcon tube where an opening had been created for feeding the bumblebees with a micropipette. After placing the bumblebee inside the falcon tube, cotton was added to keep it close to and facing the opening. The control bumblebees were fed with 10 μL of sugar water solution 50% (*w*/*v*) while the acute-exposed ones were fed with 10 μL of the imidacloprid solution. During this process, the bumblebees were continuously observed to ensure that they consumed the given amount of solution. Subsequently, they were released into the flight cages. Their flight activity was recorded for the next 96 h according to the acute exposure definition [7]. This experimental procedure was repeated three times.

### 2.5. Chronic Exposure Experiment

First, 70 bumblebees were collected from the hive for each treatment, under the red light. Bumblebee workers were carefully transferred with forceps into two mesh cages (60 × 60 × 70 cm), one for each treatment. The bumblebees stayed in the cages for 10 days and were exposed daily to low doses of imidacloprid, according to the chronic exposure definition [7]. The control bumblebees were provided with ad libitum sugar water solution 50% (*w*/*v*), while the pesticide-exposed bumblebees were provided ad libitum with the imidacloprid-treated solution. The solutions were given in 30 mL cups, filled with cotton on the top, and refilled every day. The cages were kept in a laboratory room with stable conditions (25 °C and natural light cycle). After 10 days, the bumblebees were released into the flight cages and the recording of the flying activity started. This experimental procedure was repeated twice (58 individuals per treatment in the first repetition; 55 individuals per treatment in the second repetition).

### 2.6. Data Analysis

Every time a bumblebee flew into the sensor’s measurement volume, an insect event was recorded. Since the sensor LEDs emit light at two wavelengths and the detector records in four spatial quadrants [55], an insect event might correspond to up to eight data points, depending on the insect’s flight movement (Figure 1). The aim was to record as many insect events as possible in order to acquire a high number of data points that would be used in the analysis (machine learning models, especially neural networks, are trained better with a high number of data points) [61]. Different numbers of insect events were obtained for every treatment’s repetition (See Table 1 and Table 2). Insect events were further analysed using two different methods, discussed below.

The first method used is the analysis of the feature of wbf generated from the insect events. The purpose of this analysis is to investigate whether there are immediately noticeable and quantifiable differences between the datasets generated by the pesticide-exposed and the control bumblebees. For each insect event, wbf was calculated using the wbf considered in the study by Rydhmer et al. (2022) [55]. The wbf of *B.terrestris* is 169.869 ± 11.215 (Hz) [62]; therefore, values above 270 and below 140 were considered outliers and removed before the start of the analysis since they represented errors in the algorithm. Finally, the data from each repetition were pooled together both for the acute and chronic treatment and visualized in a histogram. For each treatment the median was also calculated.

In the second part of the analysis, the intended purpose was to investigate if there were further differences between pesticide-exposed and control bumblebees that a machine learning algorithm could identify. The algorithm used is a convolutional neural network (CNN), which belongs to a class of algorithms known as supervised learning [63] and is commonly used in image and signal processing. The algorithm consists of creating two datasets that are assigned their true labels—in this case, whether or not the bumblebees had been exposed to pesticide—and subsequently “training” a model to find patterns in the data that would differentiate the two groups from each other. The model is then tested on a small percentage of yet “unseen” data to evaluate its performance on how well it learned the patterns specific to the different treatments. 

To create the datasets for training the neural network, the insect events from each repetition were pooled together by treatment type, creating an acute-exposed and a chronic-exposed dataset, which also included their respective control group data. From both datasets, a random 10% of the data were withheld to be the test set, making sure that both the pesticide-exposed and the control group were represented. Two separate models were thus trained, one to distinguish acute-exposed bumblebees from their control counterparts and one similarly for differentiating chronic-exposed bumblebees from the respective control group. 

A confusion matrix was used to evaluate the model created by the neural network. The accuracy of the model was calculated by the constructed confusion matrix according to the following formula [64]: $$Accuracy=\frac{\#\;Correct\;Predictions}{\#\;Total\;Predictions}$$

The accuracy metric and the confusion matrix help in evaluating whether the model succeeded in finding any significant, generalisable differences between the two datasets. It is, however, unknown what the sources of the differences are, which is an inherent property of a CNN such as this one [65]. To further understand and interpret the results, some of the features that the algorithm learns and how they may or may not relate to our manually computed features (such as the wbf), were visualized. In this study, a t-SNE (t-distributed stochastic neighbour embedding) was used. T-SNE is a statistical method, which projects a high-dimensional data set into a lower dimensional space [66]. In this case, the t-SNE algorithm is applied on the penultimate layer of the convolutional neural network, projecting the high-dimensional layer into a 2-dimensional map where every data point corresponds to an insect event. These two dimensions are thus “features” that the algorithm learned, but they do not have any physical or biological meaning by themselves (unlike for example the wingbeat frequency). The map that is created shows that neighbouring data points in the t-SNE map represent similar insect events. Conversely, if two data points in the map are far away from each other, the insect events are also very different. The t-SNE algorithm was applied to both trained networks. The different treatment options are shown in different colours in the t-SNE maps. This means that the more overlap, the more similar the treatment groups, and the model has greater difficulty distinguishing them. As previously mentioned, it is not known what features are actually learned by the algorithm (what is in the X and Y axes of the diagrams). To further investigate what influences the pattern that these maps show, the original t-SNE maps were colour-coded by the wbf data previously described. This would help understand if the patterns recognised by the neural network can be explained by differences in wbf.

## 3. Results

### 3.1. Acute Exposure Experiment

During the acute exposure experiment, a total of 29,517 insect events were recorded for the imidacloprid-exposed, and a total of 25,567 insect events for the control (Table 1). As mentioned in the beginning of the previous section, each insect event can result in several (2–8) data points. After removal of outliers (defined in the methods section), 184,984 data points were obtained in the imidacloprid-exposed treatment and 149,018 in the control treatment.

The median value of the wingbeat frequency of the bumblebees in the imidacloprid-exposed treatment was 178 Hz (171–186 Hz) (*n* = 184,984). In the control treatment, the median was 181 Hz (181–190 Hz) (*n* = 149,018) (Figure 2).

The confusion matrix (Figure 3) shows that the algorithm correctly recognised 2735 insect events from the acute treatment and 3433 insect events from the control. The model incorrectly predicted 1061 insect events as events of the acute treatment, while they were, in fact, events of the control and 1771 insect events as events of the control treatment while they were events from the acute treatment. The test accuracy of the model is thereby calculated to be 67%. 

The t-SNE map (Figure 4A) shows most of the insect events from both treatments (acute and control) overlapping, except for the upper right corner of the map, where mainly insect events from the acute treatment are located. Those data points from the acute treatment appear, thus, different to the algorithm from the rest.

Looking at the following t-SNE map (Figure 4B), it is noticeable that different wingbeat frequency values do not follow the same pattern observed on Figure 4B but that data points are randomly distributed instead. Therefore, the pattern observed in Figure 4B is not explained by differences in wbf. 

### 3.2. Chronic Exposure Experiment

During the chronic exposure experiment, the imidacloprid-exposed bumblebees created a total of 4580 events and the control bumblebees 7309 events in total (Table 2). This corresponded to 28,162 data points obtained in the imidacloprid-exposed treatment and 46,778 in the control treatment, after the removal of outliers (arising from non-physiological wingbeat frequency values).

The median value of the wingbeat frequency of the bumblebees in the chronic imidacloprid-exposed treatment ranged from 181 to 197 Hz with a median value of 189 Hz (181–197 Hz) (*n* = 28,162), while in the control treatment, the median value was 193 Hz (184–197 Hz) (*n* = 46,778) (Figure 5).

From the confusion matrix, the following details are obtained. The algorithm correctly recognised 1602 insect events of the chronic treatment and 2536 events from the control, while it misclassified 529 control events as chronic treatment events and 556 chronic events as control (Figure 6). The test accuracy percentage is 78%.

The t-SNE map (Figure 7A) illustrates that the data from the chronic treatment (red) insect events and control (green) are overlapping in most of the area; however, it is clear that in the red tip in the upper part of the map there are only events from the chronic treatment and, in the right area of the map, events from the control are dominant (Figure 7A). As in the acute experiment, such differences cannot be explained by the wingbeat frequency values (Figure 7B), which appear to be randomly distributed. 

## 4. Discussion

The overall aim of this study was to investigate the potential use of an optical sensor, which can automatically and in real-time, identify flying insects using machine learning to detect the sublethal effects of imidacloprid, a widely used insecticide, on *B. terrestris*. Two types of exposures were tested, acute and chronic, with field-realistic concentrations of imidacloprid. 

In both the acute and chronic treatment experiments, the wingbeat frequency (wbf) of most of the insect events recorded by the optical sensor was between 170 and 200 (Hz) (Figure 2 and Figure 5). This agrees with a previous study which reported that the wingbeat frequency of *B. terrestris* is 169.869 ± 11.215 (Hz) [62]. Therefore, our results demonstrate that the flight activity of *B. terrestris* inside the flight cages generated flight events with wbf in line with the literature, validating the data recorded by the optical sensor. To investigate the ability of the sensor to differentiate events between acute-exposed and control bumblebees, a neural network was trained with the insect events of both treatments. The calculated accuracy of the model was found to be 67%, which means that from the “unseen” insect events that the algorithm was tested on, 67% were classified to the correct category. This reveals a difficulty for the model to accurately identify the bumblebees of the acute treatment. Within the incorrect predictions of the model in the confusion matrix, the model misdiagnosed more insect events as control (1771) than as acute-treated (1061), meaning that the model is more discreet towards the acute insect events. The t-SNE map of the acute and control insect events (Figure 4A) showed that some events in the acute treatment were separated from those in the control treatment. Yet, those differences were not visible in the colour-coded t-SNE map overlaying the wingbeat frequency of the insect events between the two treatments (Figure 4B). Therefore, the differences that the algorithm sees cannot be explained by the wbf. Given the fact that a deep neural network (a black box method) [65] was used to train this model, it is not surprising that the differences that the algorithm sees are difficult to trace. In other words, the recorded insect events likely included features of the insect flight that remain unknown. These features might be influenced by the pesticide exposure; thus, the algorithm can pick up those differences. This is why the t-SNE map of acute and control insect events presents few differences between the treatments which are not explained from the colour-coded wbf maps. A similar pattern is observed for the chronic exposure experiment. The test accuracy obtained for this model was 78%, meaning that the algorithm was able to correctly categorise 78% of the events registered by the sensor. This is confirmed by the t-SNE map (Figure 7A), where the model distinguishes many of the control events from the chronic-exposed events. However, as with the acute exposure experiment, the colour-coded t-SNE maps of the wing beat frequency (Figure 7B) do not seem to influence the pattern seen in the first t-SNE map.

It is important to mention that the data that the algorithm was trained on was very similar; thus, high accuracy performance values of the model were not expected. The insect events look similar for two reasons: First, all the insect events are from the same insect species so many of the features attributable to the bumblebees’ physical characteristics (body size, wing size, melanisation) are similar. Second, the pesticide concentration that the bumblebees were fed with was low; thus, its impact might be minimal. The differences that allow the algorithm to distinguish the flight events between the non-exposed and the imidacloprid-exposed bumblebees might be due to differences in behaviour. For instance, a previous study showed that acute imidacloprid exposure can affect flight endurance, with imidacloprid-exposed bumblebees flying with higher velocity and terminating their flight faster [41]. Another study also found that chronic exposure to imidacloprid on bumblebees caused increased movement speed but decreased the time of their activity [43]. Imidacloprid exposure has also been found to have an effect on bumblebee sonication performance. Bumblebees that consumed imidacloprid were observed to sonicate significantly less than bumblebees that did not [35]. 

The model that was trained with the events of chronic and control treatment (78% accuracy) achieved better accuracy than the model that was trained with the acute and control treatment events (68% accuracy). Even though it remains unknown which features the algorithm considers, this indicates that the effects that the algorithm detected could be related to the length of exposure. In fact, this aligns with another study in which minimal impacts of acute exposure to thiamethoxam were detected while chronic exposure significantly impaired learning and memory ability of the bumblebees [67]. Chronic exposure tests show the long-term effects of daily exposure to a pesticide, allowing for more field-realistic insights while acute exposure (single dose) tests provide first evidence of the real risk of a pesticide [7]. Bumblebees are more likely to face long-term exposure (chronic exposure) to pesticides through direct contact during foliar spray applications [18]; contact with nectar, pollen, and guttation fluid (xylem sap) [46]; through drinking contaminated water; nesting in contaminated soil; or from drifting dust that seed coatings release while sowing [47]. Toxicity depends on the exposure route; oral exposure is considered more toxic than contact exposure [57]. For social species, exposure can also occur through contaminated hive materials. Bees might bring back contaminated nectar and pollen to store in the hive to feed the queen and the larvae [47].

Over the past few years, there has been an increasing emphasis on studying the impacts of sublethal doses of imidacloprid on bees [68,69,70,71,72]. These studies have consistently highlighted the existing knowledge gaps concerning inadequate modelling and monitoring of risk assessment regulations, as well as the limited understanding of pesticide exposure in real-field conditions [68,70,73,74]. To address these gaps, researchers have developed models to simulate bee population dynamics, primarily focusing on honeybees and pesticide toxicity [72,75,76]. Despite this, to improve the accuracy of predictions made with these models, scientists stress the critical importance of improving field data collection on pesticide exposure [72,75,76]. Tools that offer automatic, long-term field data collection have the potential to bridge the existing gaps in field data in a fast and cost-effective way. Bee counters [77] and video tracking software [78] and other types of sensors [79,80] are tools such as are available providing valuable data for pesticide effects, but they are primarily designed for beehives, limiting their applicability to other species. The development of more versatile tools is essential to capture data from a broader range of species and environments, facilitating more comprehensive research and insights into pesticide exposure and its impact on various pollinators. The tool described in this study can provide non-invasive and long-term data on pollinator health and could be useful for monitoring and studying pollinator populations in the field, allowing for timely interventions and conservation efforts to protect pollinator species. However, these results are based on laboratory experiments and further experiments need to be carried out under semi-field and field conditions. Many studies argue that the sublethal effects of pesticides on bees are not consistent between lab and field studies [81,82,83]. One study suggests that this is due to the fact that the “field-realistic” concentrations of pesticides used in lab studies are higher than in the field [81], whereas another study found higher concentrations of pesticides in bumblebee nectar [11] than the ones reported as “field-realistic” in the literature [84]. This study exposed bumblebees only to the “field-realistic” concentration according to the literature on imidacloprid. Treatments with different concentrations of imidacloprid might also give interesting and useful results and should be considered for future studies [35,83]. 

Overall, our results show that there is potential for the use of optical sensors to investigate the behavioural effects of pesticides on bumblebees and other pollinators. Sublethal effects are generally harder to identify, and some could be subtle enough to evade detection from traditional means of analysis. As demonstrated here, the addition of machine learning algorithms could have wide-ranging implications for our understanding of previously undetected effects of pesticides and other harmful substances on flying insect behaviour. 

## 5. Conclusions

This study shows the potential use of an optical sensor as a tool for detecting the sublethal effects of pesticides in bumblebees under laboratory conditions. Our results demonstrate that an optical sensor was able to detect differences in flight signals of *B. terrestris* workers after chronic exposure to imidacloprid with 78% accuracy. Further studies could develop the optical sensor as a monitoring tool in the field, where the pollinators’ activity could be continuously monitored and assessed in real-time and combined with data on the presence of other stressors. The use of the optical sensor is a fast, automated, cost-effective, and standardised method to obtain information on sublethal effects that would otherwise be overlooked. The development of such a monitoring tool will lead to more effective management and conservation of pollinators.

## Figures and Tables

**Figure 1 insects-14-00713-f001:**
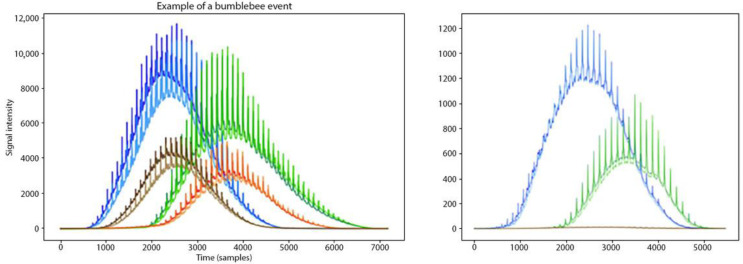
Two bumblebee events recorded from the sensor. In the left bee event, the bee flew across the 4 quadrants of the photodiodes, and due to the 2 wavelengths of the sensor, 8 data points correspond to this bee event. In the right bee event, the bee flew only across the 2 quadrants of the photodiodes, so 4 data points correspond to this bee event.

**Figure 2 insects-14-00713-f002:**
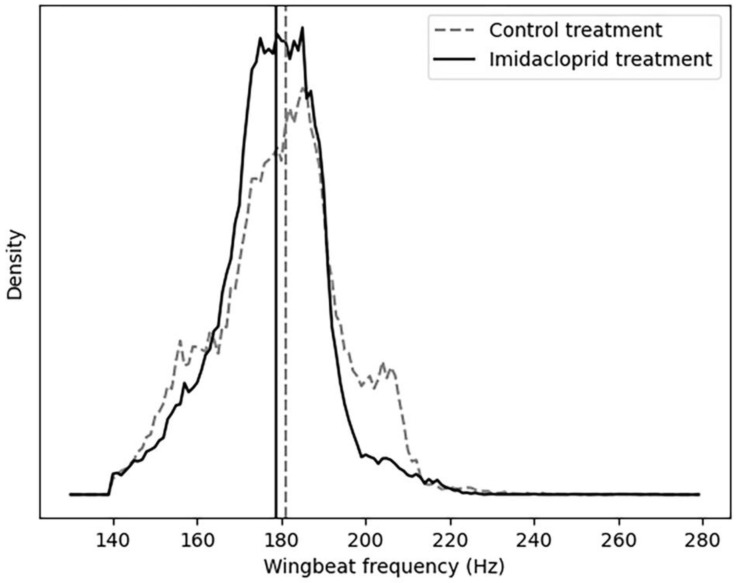
Wingbeat frequency of the bumblebees (control bumblebees = dash line; treated bumblebees = solid line) flying in the flight cages during the acute exposure experiment. All events from all repetitions were pooled together.

**Figure 3 insects-14-00713-f003:**
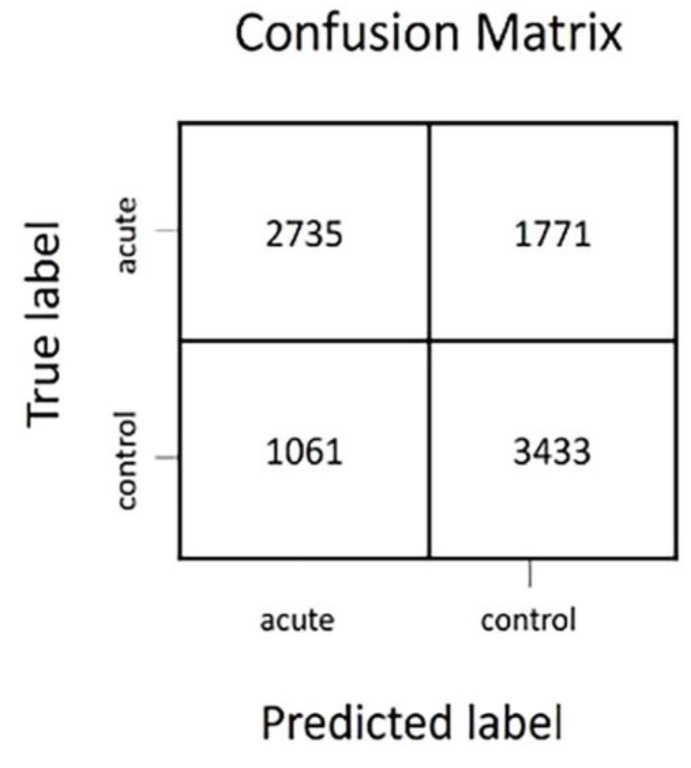
Confusion matrix for imidacloprid-exposed (acute) and control insect events.

**Figure 4 insects-14-00713-f004:**
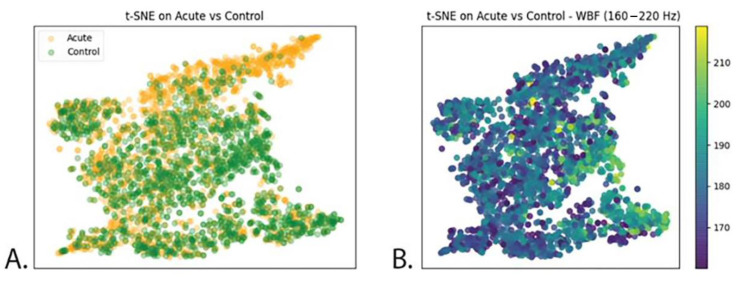
(**A**): t-SNE map representing all insect events recorded. Orange dots = imidacloprid treatment. Green dots = control treatment. (**B**): t-SNE map with wingbeat frequency of the bumblebees overlaid as a colour gradient on top of the data. Wingbeat frequency scale ranges from 160 Hz to 220 Hz.

**Figure 5 insects-14-00713-f005:**
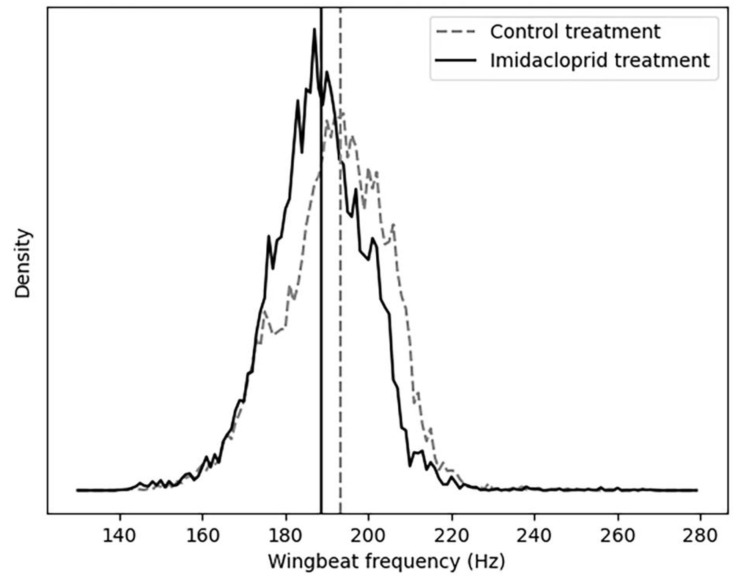
Wingbeat frequency of the bumblebees (control bumblebees = dash line; treated bumblebees = solid line) flying in the flight cages during the chronic exposure experiment. All events from all repetitions were pooled together.

**Figure 6 insects-14-00713-f006:**
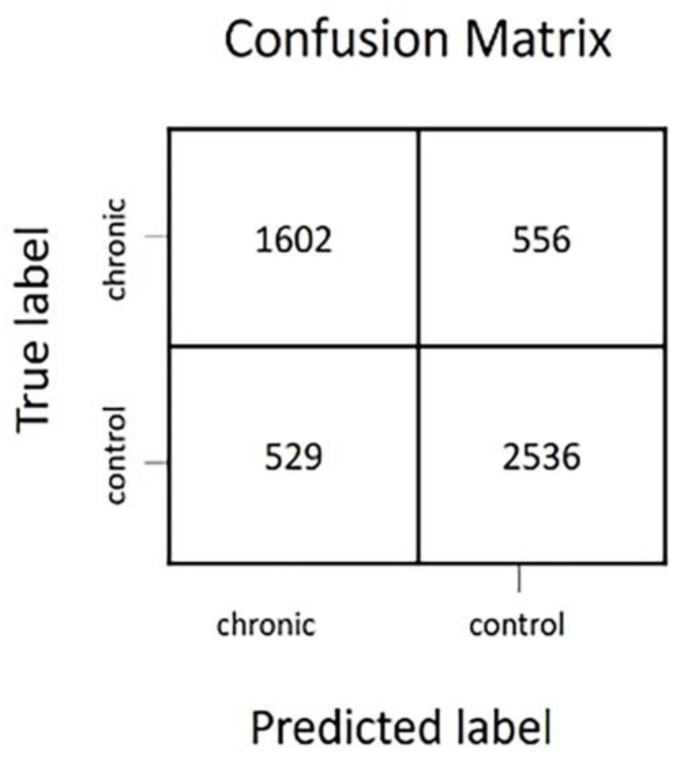
Confusion matrix for imidacloprid-exposed (chronic) and control insect.

**Figure 7 insects-14-00713-f007:**
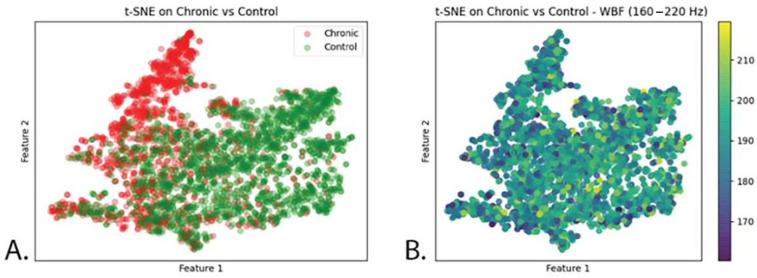
(**A**): t-SNE map representing all insect events recorded. Red dots = imidacloprid treatment. Green dots = control treatment. (**B**): t-SNE map with wingbeat frequency of the bumblebees overlaid as a gradient on top of the data. Wingbeat frequency scale ranges from 160 Hz to 220 Hz.

**Table 1 insects-14-00713-t001:** Number of recorded bumblebee flight events for each repetition and treatment during the acute exposure experiment.

Repetition	Treatment	Number of Insect Events	Number of Data Points
1	imidacloprid-exposed	6799	41,047
	control	8449	50,855
2	imidacloprid-exposed	15,399	84,587
	control	8570	47,487
3	imidacloprid-exposed	7319	59,350
	control	8548	50,676

**Table 2 insects-14-00713-t002:** Number of recorded bumblebee flight events for each repetition and treatment during the chronic exposure experiment.

Repetition	Treatment	Number of Insect Events	Number of Data Points
1	imidacloprid-exposed	2057	11,789
	control	3712	24,224
2	imidacloprid-exposed	2523	16,373
	control	3597	22,554

## Data Availability

The data presented in this study are available upon reasonable request from the corresponding author.

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
