# Peer review of "A New Approach for Detecting Sublethal Effects of Neonicotinoids on Bumblebees Using Optical Sensor Technology"

_insects, 2023, doi:10.3390/insects14080713_

Round 1

Reviewer 1 Report

I have been asked to review the study by Chatzaki et al., " A New Approach for Detecting Sublethal Effects of Neonicotinoids on Bumblebees Using Optical Sensor Technology".

The research is very interesting and gives a modern approach to the study of pesticides effects in pollinators, particularly bumblebees.

The research is well conducted, however I found the manuscript writing a bit confusing in some points and not easy to immediately understand. Particularly, I would suggest to rewrite some paragraphs of the MM section and discussion trying to give it a more schematic and concise cut.

Also, I believe that the true value of the described tool is to be found in its use in research to assess possible unknown effects of pesticide use, also in a perspective of analyzing elements which would not be easy for humans researchers. Therefore, it could be important to stress the applicability of this tool.

I have also some minor concerns which are listed below.

Line 29-30: source of exposure should be defined in the abstract

Line 59: it would be appropriate to change the sentence as “Considering the findings…”, moreover it would be nice to write a few words about how the test is carried out.

Line 65: paralysis is not the only clinical symptom of neonicotinoid exposure. I would suggest reporting other symptoms which could strengthen the importance of the study

Line 65-69: Consider changing the sentence to: “They are systemic insecticides which are applied to plants as foliar sprays, and to soil as drenches, in granular form, or as seed coating. Seed coating is the most common application method of neonicotinoids as their systemic action offers protection to the plant as it grows [14,16], however only around 10% is absorbed by the plant while the rest remains in the soil”.

Line 74-75: References 19-21 were published in year 2012, I believe there are more recent studies or even better reviews that could be added to the bibliography. I would also suggest to update bibliography throughout the manuscript. Moreover, I would change the word “risk” with “effects” as references 19-21 focus on the effects of exposure and not the risk of exposure.

Line 81-83: Consider changing the sentence as follows: “Since the moratorium, crop yields have decreased, while pest control necessities have increased, and neonicotinoids were replaced with other insecticides, which are not as effective but can still harm pollinators.”

Lines 86-91: Probably it would increase readability to move them to line 80 after reference 26.

Line95-97: You could probably remove this sentence to improve readability of the manuscript, or move the reference to line 141 and line 166 next to reference 57, where the administered dose of pesticides is calculated.

Lines 112-115: Consider changing the sentence as follows: “Most of the up-forementioned studies were conducted under laboratory conditions; therefore, to have  a complete picture of the exposure effects of neonicotinoids on bees more research is needed especially for field exposure.”

Line 119: “automatic traps [50], and cameras [51], radars [52,53]…”

Line 120: “The optical sensor described in Rydhmer et al.”, the year of the study is missing. Check and harmonize throughout the manuscript the addition or removal of the year of publication after the name of authors.

Line 122: “volume[55].” Add space

Line 122-125: Consider changing the sentence as follows: “The recorded signals generate insect data which is analyzed using different methods, by evaluating features of the insect’s signal (for example the wingbeat frequency, body size, wing size) or by machine learning.” However, it is not clear how the evaluation of the features and how machine learning analysis are performed. I would suggest to add a sentence to better clarify these elements.

Line 129: “The optical sensor is a real-time….”

Line 140-142: Consider changing the sentence as follows “Bees were orally exposed to field realistic concentration of imidacloprid, which corresponds to 10μg / L (10ppb) [46,57] and chronic and acute exposure were tested.”

Line 148: were the hive boxes kept in an incubator, or how were standard conditions maintained? Was there also a standard humidity?

Line 150-152: Consider changing the sentence as follows “For each repetition bees were collected from different hives, and from each hive individuals of different sizes were sampled to avoid size-biased data since wing beat frequency (wbf) can be affected by the size of the body [58].”

Line 157-168: This paragraph is not easy to understand. I would suggest to rewrite the paragraph in order to increase readability.

Line 182: I would suggest to use the specific “bumblebee” word throughout the manuscript when it applies to reinforce to context of the study to avoid possible confusion.

Line 192: “15 ml falcon tube”

Line 197: “Subsequently, bees were released…”

Lines 208-209: Feels like repetition.

Lines 200-212: The following sentence “After 10 days, an equal number of bees from both treatments was released into the flight cages (58 in the first repetition and 55 in the second repetition), where the recording of the flying activity started. This experimental procedure was repeated two times” is a bit confusing. I would suggest Authors to try to be more concise and specific.

Line 232: Consider changing the sentence as follows “For each insect event wbf was calculated using the wbf considered in the study by Rydhmer et al. [55].”

Line 266: the sentence “Several methods exist for creating such visualisations” can be removed.

Line 421 and line 423: (78% accuracy) and (68% accuracy).

Line 425: “In fact, this aligns with another study in which minimal….”

Line438-439 and 453-457:  It is difficult to imagine to use optical sensors as a monitoring tool to “diagnose” poisoning of bumblebees. On the contrary, I believe that this study sets basis for the use of optical sensors as a scientific tool to better understand the effects of pesticides on pollinators. I would be more prone to discuss results in this sense.

I suggest to adjust some points to improve readability of the Manuscript. I have already proposed some changes in the general review but I might have missed some.

Author Response

Thank you for your valuable comments. Please see the attachment.

Reviewer 2 Report

insects-2492672 Reviewer comments

Manuscript insects-2492672: A New Approach for Detecting Sublethal Effects of Neonicotinoids on Bumblebees Using Optical Sensor Technology

The manuscript is very interesting. The authors investigated the potential use of this sensor as a tool for monitoring the sub-lethal effects of pesticides on bees. Bombus terrestris workers were exposed to field-realistic doses of imidacloprid. Two types of exposures were tested, acute and chronic. The flight activity of pesticide-exposed and non-exposed bees was recorded and the events of the insect flights recorded by the sensor were used in two ways: to extract the values of the wingbeat frequency and to train machine-learning models. The results showed that the trained model was able to recognize differences between the events created by pesticide exposed bees and control bees.

The uniqueness of the text is 90% by antiplagiarism.net

The experimental and statistical methods are correct.

The English is almost good but there are some misspellings.

There are some mistakes and comments:

Lines 58, 59 - after the sentence - However several studies have shown that pesticide toxicity varies across species [9,10]. - add citation - (Lim et al., 2020). - add to the references - Lim, S.; Yunusbaev, U.B.; Ilyasov, R.A.; Lee, H.S.; Kwon, H.W. Abdominal contact of fluvalinate induces olfactory deficit in Apis mellifera. Pesticide Biochemistry and Physiology 2020, 164, 221-227, doi:10.1016/j.pestbp.2020.02.005.

Line 92 - rewieved - should be - reviewed.

Line 134 - aquired - should be - acquired.

Line 410 - melanisation - should be - melanization.

Lines 413, 415 - imidaclroprid - should be - imidacloprid.

Discussion part is weak, please add more discussion with comparison with other studies.

Please improve the manuscript according to the above comments.

The English is almost good but there are some misspellings.

Author Response

(The authors gave the same response as above.)
